# DNAJC12 Deficiency, an Emerging Condition Picked Up by Newborn Screening: A Case Illustration and a Novel Variant Identified

**DOI:** 10.3390/ijns10040074

**Published:** 2024-11-19

**Authors:** Tsz Sum Wong, Sheila Suet Na Wong, Anne Mei Kwun Kwok, Helen Wu, Hiu Fung Law, Shirley Lam, Matthew Chun Wing Yeung, Toby Chun Hei Chan, Gordon Leung, Chloe Miu Mak, Kiran Moti Belaramani, Cheuk Wing Fung

**Affiliations:** 1Department of Paediatrics and Adolescent Medicine, Princess Margaret Hospital, Hong Kong, China; wts0752@ha.org.hk; 2Department of Paediatrics and Adolescent Medicine, Hong Kong Children’s Hospital, Hong Kong, China; wongsn2@ha.org.hk (S.S.N.W.); kwokmk@ha.org.hk (A.M.K.K.); helen.wu@ha.org.hk (H.W.); hf.law@ha.org.hk (H.F.L.); lw018@ha.org.hk (S.L.); 3Newborn Screening Laboratory, Department of Pathology, Hong Kong Children’s Hospital, Hong Kong, China; ycw186@ha.org.hk (M.C.W.Y.); cch191@ha.org.hk (T.C.H.C.); makm@ha.org.hk (C.M.M.); 4Division of Genetic and Genomic Pathology, Department of Pathology, Hong Kong Children’s Hospital, Hong Kong, China; lkc344@ha.org.hk

**Keywords:** DNAJC12 deficiency, hyperphenylalaninemia, newborn screening, Hong Kong

## Abstract

DNAJC12 deficiency is a recently described inherited metabolic disorder resulting in hyperphenylalaninemia and neurotransmitter deficiency. The effect of treatment on the prevention of neurological manifestations in this newly reported and heterogenous disorder is not fully understood, and the optimal treatment strategy remains to be elucidated. The global or regional incidence of the disease is yet to be estimated. Here, we report the first individual diagnosed with DNAJC12 deficiency in Hong Kong; the condition was picked up by newborn screening due to hyperphenylalaninemia after ruling out phenylalanine hydroxylase deficiency and other tetrahydrobiopterin related disorders. Compound heterozygous variants in the *DNAJC12* gene were identified, which included a novel missense change and a nonsense pathogenic variant. Treatment with neurotransmitter precursors (tetrahydrobiopterin, levodopa, and oxitriptan) was initiated at four months of age, and dietary protein restriction was started at four years and six months of age. He remains asymptomatic at four and a half years of age, apart from having mildly impaired socio-communication and language development. In this report, we discuss the current diagnostic approach to hyperphenylalaninemia in newborn screening and the uncertainties that exist in the clinical outcome from earlier detection, treatment, and monitoring of DNAJC12-deficiency patients.

## 1. Introduction

In the 1960s, Dr. Robert Guthrie pioneered newborn screening by developing a test to detect elevated phenylalanine (Phe) concentrations on dried blood-spot cards, which could be collected shortly after birth to screen for phenylketonuria (PKU) [1]. Although tandem mass spectrometry has become the mainstay of newborn screening nowadays and allows the analysis of multiple analytes, an elevated Phe (hyperphenylalaninemia (HPA)) remains one of the primary analytes to initiate call backs in newborn screening programs worldwide.

PKU, also known as phenylalanine hydroxylase (PAH) deficiency, has been a core newborn screening target in most programs. This condition is particularly prevalent in the Caucasian population [2]. Early initiation of dietary intervention and sapropterin dihydrochloride (BH_4_) for BH_4_ responsive cases may prevent adverse neurological outcomes if Phe levels can be maintained below toxic levels. Although European and American guidelines differ in their treatment thresholds for patients over 12 years of age, both recommend initiating treatment for patients aged 12 years or younger if plasma Phe levels exceed 360 μmol/L [3].

HPA has also been identified in BH_4_ metabolism disorders such as 6-pyruvoyl-tetrahydropterin synthase (PTPS) deficiency, autosomal recessive (AR) guanosine-5′-triphosphate cyclohydrolase I (GTPCH) deficiency, and dihydropteridine reductase (DHPR) deficiency. BH_4_ metabolism disorders account for approximately 1–2% of cases with HPA in most populations [4,5,6]. However, in Asian populations, the risk to an individual with HPA having a BH_4_ metabolism disorder is even higher (8.55–30%), with PTPS deficiency being the most common condition [7,8,9]. Therefore, although BH_4_ metabolism disorders are considered secondary target conditions in the recommended uniform screening panel, PTPS deficiency is a primary target condition in the newborn screening program of Hong Kong (HK).

However, in a minority of cases, the causes of HPA remain unresolved [4,5,6]. In 2017, a novel condition named DNAJC12 deficiency, associated with HPA, was reported. This condition results from bi-allelic variants of the DNAJC12 gene, which encodes the co-chaperone of PAH and other aromatic amino acid hydroxylases [10]. DNAJC12 interacts with and prevents the misfolding of various proteins, including PAH, tyrosine hydroxylase, and tryptophan hydroxylase. Consequently, in addition to HPA, patients with *DNAJC12* biallelic variants may suffer from a deficiency of multiple neurotransmitters, such as dopamine and serotonin.

The understanding of the prevalence, clinical presentation, natural history, and prognosis of patients with DNAJC12 deficiency remains limited, with diverse clinical phenotypes reported in the literature. The clinical manifestations of reported cases have ranged from being asymptomatic or experiencing mild neurological involvement to progressive dystonia and intellectual disability [10,11,12,13,14]. Also, there is no established consensus on the treatment and monitoring of DNAJC12 deficiency. Current treatment strategies are largely based on the consensus guideline published for disorders of BH_4_ deficiencies [9]. The role of sapropterin dihydrochloride and neurotransmitter replacement therapy in preventing neurological manifestations is still not fully understood.

In this report, we present the first documented case of DNAJC12 deficiency-related HPA in Hong Kong, along with identification of a novel variant in the *DNAJC12* gene. This case underscores the need to reevaluate the diagnostic approach to HPA and highlights the existing uncertainties in the treatment and monitoring of DNAJC12-deficiency patients.

## 2. Case Report

The proband was a boy born at term via normal vaginal delivery to non-consanguineous parents. His mother is of Chinese descent, and his father is Spanish. He enjoyed an uneventful perinatal history, and the family history was unremarkable.

The dried blood spot (DBS) collected on day two of life for newborn screening (NBS) was flagged for HPA. The Phe level was 233 μmol/L (reference range: 34–84 μmol/L), and there was an elevated Phe-to-tyrosine ratio of 2.38 (reference range: 0.27–1.1). The subsequent DBS sample collected on day 12 of life showed a persistent elevation at 236 μmol/L (reference range: 24–70 μmol/L). Plasma Phe collected on the same day was also elevated at 272 μmol/L (reference range: 49–107 μmol/L). Prolactin (PRL) level was raised at 2526 mIU/L (reference range: 78–1721 mIU/L). Urine pterins did not reveal any neopterin or biopterin elevations by liquid chromatography with tandem mass spectrometry (LC-MS/MS). Therefore, the likelihood of BH_4_ metabolism disorders was considered low, and genetic testing was targeted for the *PAH* gene only. However, no clinically significant variants were detected in the *PAH* gene by Sanger sequencing. In the absence of a heterozygous *PAH* variant, the possibility of PAH deficiency was deemed less likely, and thus, Multiplex ligation-dependent probe amplification (MLPA) assay to identify deletion and duplication gene defects was not performed.

The HPA persisted with a plasma Phe level of 312 μmol/L (reference range: 52–116 μmol/L) at one month of life. Further investigation to rule out BH_4_ metabolism was performed. Urine pterins were repeated and came back normal. Analysis of DHPR activity was normal at 2.4 mU/mg Hb (reference normal >1.1) at three months of age. A lumbar puncture performed at three months of life revealed low cerebrospinal fluid (CSF) homovanillic acid (HVA) 248 nmol/L (reference range: 295–932 nmol/L) and 5-hydroxyindolacetic acid (5-HIAA) 75 nmol/L (reference range: 114–336 nmol/L). CSF Phe was elevated at 42 μmol/L (reference range: 5–21 μmol/L).

At three months of age, he also underwent a targeted analysis of the whole *DNAJC12* gene (NM_021800.3) by Sanger sequencing. All the coding exons and the flanking introns (± 40 bp) of the gene were analyzed. He was found to be compound heterozygous for two variants, NM_021800.3: c.214C>T p. (Arg72*) and NM_021800.3: c.185C>A p. (Ala62Glu). Parental genotyping showed that the nonsense variant is inherited from the father, while the missense variant is inherited from the mother.

The nonsense variant, c.214C>T, replaced an arginine with a termination codon at position 72 of the DNAJC12 protein. This variant is a known pathogenic variant that has been reported in multiple patients with DNAJC12 deficiency in the homozygous state [15].

The missense variant, c.185C>A, is a novel variant that is absent in the control database (gnomAD exomes and genomes v2.1.1). The REVEL score was 0.808, indicating moderate pathogenicity. The clinical presentations of isolated HPA, normal urine and CSF pterin patterns, and CSF findings of reduced 5-HIAA and HVA are highly suggestive of DNAJC12 deficiency. Therefore, the missense variant was curated as likely pathogenic according to the American College of Medical Genetics and Genomics (ACMG) standards and guidelines for the interpretation of sequence variants [16,17,18,19].

Although the patient had been asymptomatic and plasma Phe levels were below 360 μmol/L before four months of age, the CSF findings showed neurotransmitters deficiency. In view of the genetic findings, we adopted a management approach to aim at maintaining Phe levels below 360 μmol/L and replacing the deficient neurotransmitters. Consequently, therapy with sapropterin dihydrochloride, the synthetic form of the cofactor tetrahydrobiopterin (BH_4_), at 1.3 mg/kg/day; levodopa/carbidopa at 1.3 mg/kg/day; and 5-hydroxytryptophan at 1 mg/kg/day was initiated. The patient continued on a normal unrestricted diet.

Despite gradual escalation of drug dosages, the plasma Phe levels never normalized (Table 1). At one year of age, the Phe levels began to intermittently rise above 360 μmol/L, reaching a maximum of 589 μmol/L at 16 months of age. Reassessment of CSF neurotransmitters at that juncture failed due to unsuccessful sedation for lumbar puncture. Escalation of sapropterin dihydrochloride dosage up to 20 mg/kg/day made the patient able to reach the Phe target most of the time, subsequently, but at 21 months of age, his plasma Phe levels rose markedly again to 567 umol/L [9]. Although he had no dystonia or involuntary movements, speech therapist assessment revealed mild speech delay, especially expressive language impairment. He also had fair eye contact, with decreased social responses. Therefore, lumbar puncture was re-attempted at 21 months of age to assess treatment adequacy.

The CSF HVA and 5-HIAA, this time, were within the lower end of the reference range (at 369 nmol/L and 137 nmol/L, respectively), suggesting that central dopamine replacement was adequate. This did not correlate with the high plasma PRL level of 1268 mIU/L (49–280 mIU/L) measured concomitantly. It was postulated that the elevation of plasma PRL was due to sampling issues from difficult venipuncture. Therefore, blood sampling was performed via a large bore intravenous cannula after allowing the patient to rest after cannula insertion. Of note, although the patient was still mildly irritable when blood was drawn from the cannula, his plasma PRL dropped to 469 mIU/L with this sampling method, before any further escalation of levodopa dosage.

After 21 months of age, the patient’s plasma Phe levels were controlled to less than 360 umol/L, most of the time, with sapropterin dihydrochloride. Unfortunately, after four years and three months of age, plasma Phe levels rose above 500 umol/L at multiple occasions, despite maximizing sapropterin dihydrochloride dosage at 20 mg/kg/d. Dietary restriction was implemented at four and a half years of age by limiting total daily protein intake to 30 g per day (1.7 g/kg/day), resulting in a drop in plasma Phe level to below 300 umol/L.

At the time of the writing of this report, the patient was four and a half years old. The formal developmental assessment conducted by the Child Assessment Centre at three years and seven months of age found his language expression to be equivalent to children of two years and nine months of age and that he had traits of Autism Spectrum Disorder (ASD). The diagnosis of ASD was confirmed by a child psychiatry assessment at four years and six months of age. Otherwise, he demonstrated age-appropriate cognition, language comprehension, fine motor abilities, and gross motor abilities. He has been receiving pre-school rehabilitation services since starting kindergarten at three years and nine months of age.

## 3. Discussion

Here, we report the first case of DNAJC12 deficiency diagnosed in Hong Kong, identified through newborn screening due to HPA. Although our patient is from a mixed ethnic background (Chinese and Spanish), it can be concluded that pathogenic variants in the *DNAJC12* gene do exist in the Chinese population. In fact, there are three other Chinese cases reported in the literature at the time of writing up this report [12,20,21]. Table 2 summarizes the pathogenic and likely pathogenic variants of the *DNAJC12* gene in the Chinese patients reported to date. The variant inherited from our patient’s Chinese mother has not been reported to date in other population controls’ exomes and genomes. Considering that multiple countries, including Spain, Italy, Middle Eastern countries, and China, have described cases of DNAJC12 deficiency, this disease may well be pan-ethnic [10,12,13,20,21,22]. Further epidemiological studies are needed to determine its burden in the Chinese population and worldwide.

There are no data in the literature regarding the prevalence or incidence of DNAJC12 deficiency. Through the newborn screening for inborn errors of metabolism (NBSIEM) program in Hong Kong, we have identified one case of DNAJC12 deficiency (our case), five cases of PKU, and two cases of PTPS deficiency due to HPA between October 2015 and December 2022. A total of 125,688 newborns born in the eight birthing hospitals within the public healthcare system of Hong Kong were recruited to the NBISEM program during this period. Based on these data, the estimated incidence rate of DNAJC12 deficiency in Hong Kong was 0.8 per 100,000 live births. However, considering that only one case has been identified by NBS so far in Hong Kong, and that cases may present in older age groups based on reports in the literature, more longitudinal data are required to better estimate the prevalence and incidence of this disorder.

Previously, the diagnostic algorithm of HPA involved biochemical investigations and molecular diagnostics to differentiate PAH deficiency and BH_4_ metabolism disorders, including the use of dried blood spots and urine for pterins analyses [9]. However, recent reports suggested that *DNAJC12* genotyping should be performed in all cases of HPA that have unrevealing findings on the *PAH* gene and BH_4_ metabolism disorders. Blau et al. emphasized the importance of ruling out DNAJC12 deficiency in patients with unexplained HPA and proposed a diagnostic flowchart of HPA in their report [23]. A recent article published by Gallego et al. in 2020 reported that 20 out of 50 previously unresolved HPA cases were eventually found to have biallelic variants of *DNAJC12* by retrospective genetic analysis [13]. Moreover, 3 of these 20 cases were carriers of a pathogenic variant in the *PAH* gene but were previously misdiagnosed to have PAH deficiency [13]. Therefore, for all cases of HPA, *DNAJC12* gene testing should be considered, either tested together with the *PAH* gene and BH_4_ metabolism disorders or as soon as these disorders have been ruled out. Diagnostic algorithms for HPA were proposed by Blau et al. and the ACMG newborn screening ACT sheets and algorithm [23,24].

Conversely, additional research is necessary to ascertain the false-negative rate of NBS in detecting DNJAC12 deficiency. We have not encountered any false-negative cases to date in our NBS program in Hong Kong, where the surveillance for false-negative cases requires voluntary reporting from pediatrics specialists and physicians. However, in the literature, at least three cases with available NBS Phe levels were reported to have normal results, yet were later diagnosed to have DNAJC12 deficiency during the workup of neurological presentations [10,11,25]. Given that patients with DNAJC12 deficiency can be asymptomatic, it is challenging to estimate the number of cases missed by NBS. Nonetheless, clinicians should be aware that a normal NBS Phe level does not exclude this potentially treatable condition if clinical symptoms suggest the disorder.

Unfortunately, the limited literature available regarding the clinical course and natural history of DNAJC12 deficiency makes it challenging to recognize these patients clinically due to the marked variability in severity and presentations, ranging from mild developmental delay to severe dystonia and intellectual disability. In the initial report of six patients by Anikster et al. in 2017, all cases exhibited severe neurological symptoms if not treated with sapropterin dihydrochloride and neurotransmitter precursors [10]. Subsequent reports had expanded the clinical spectrum with cases of milder phenotypes [11,12,13,14]. The 20 cases of DNAJC12 deficiency reported by Gallego et al. were either mildly affected or entirely asymptomatic [13]. Out of these 20 patients, only 1 had dietary Phe restriction during pregnancy, 1 received neurotransmitter precursors treatment and dietary Phe restriction, and 2 were treated with sapropterin dihydrochloride. Despite the lack of treatment for most of this cohort, which included adult patients up to the age of 40 years, only four patients had constant neurological manifestations, including speech delay, intellectual disability, attention difficulties, autistic features, oculogyric crisis, and seizures. The rest of the cohort members were documented to be asymptomatic and free of neurological involvement. Thus, optimal modalities and timing for treatment for DNAJC12 deficiency remain debatable, especially for those with milder phenotypes. Nonetheless, timely diagnosis of DNAJC12 deficiency is important to ensure early treatment with sapropterin dihydrochloride and neurotransmitter precursors, which have been reported to prevent adverse neurological outcomes in those with severe phenotypes [10].

In our patient, treatment was initiated at four months of age. To our knowledge, only one patient in the literature reported by Anikster et al. had treatment (neurotransmitters, BH_4_, and folinic acid) initiated at a younger age (three months) than our patient [10]. This patient had normal clinical examinations at 22 months old, while our patient exhibited mild speech delay and traits of ASD [10]. Speech delay and behavioral issues such as ASD and Attention Deficit Hyperactivity Disorder have been described in multiple DNAJC12 deficiency patients in the literature [10,14]. Whether these clinical manifestations were incidental occurrence or features of the disorder cannot be completely ascertained, and their possible relationship with the delay in treatment initiation cannot be ruled out. Before more long-term research is available to study the clinical course of DNAJC12 deficiency, early diagnosis and treatment should be the goal in managing these patients. Checking the plasma or dried blood-spot Phe level in patients with behavioral disorders can be considered if no obvious causes have been identified.

The consensus guideline published for disorders of BH_4_ deficiencies advised against unnecessary dietary restrictions if Phe level can be controlled [9]. For our patient, the Phe target of 360 umol/L was met most of the time before four years of age, with sapropterin dihydrochloride treatment. Daily protein restriction was deemed necessary after four years of age due to high plasma Phe levels, achieving improved Phe control. Although the overall impact of dietary protein or Phe restriction is yet to be determined in patients with DNAJC12 deficiency, it is likely an essential adjunct to sapropterin dihydrochloride when the Phe target cannot be reached. Around a quarter of reported patients in the literature were treated with dietary restriction [22].

Regarding treatment monitoring of BH_4_ metabolism disorders, CSF analysis could be helpful, while plasma PRL is a more easily obtainable biomarker. Although it is non-invasive and can serve as a marker to reflect the adequacy of central dopamine replacement, the limitation of plasma PRL concentration lies in its physiological fluctuation with emotional or physical stress, physical activity, and the circadian rhythms, making it an unreliable marker to monitor treatment [9,26]. In fact, some reviews and clinical practice guideline have suggested that the stress or struggling associated with venipuncture, which is common in young patients, may cause false elevation in PRL levels [27]. The reduction of plasma PRL levels after using a large bore intravenous cannula for sampling in our patient reiterated this point. Therefore, caution should be exercised when utilizing plasma PRL levels to guide medication titration [9].

To further complicate matters, CSF neurotransmitters may not be appropriate markers to indicate clinical severity, as low CSF HVA and HIAA have been reported in asymptomatic or mildly affected cases of DNAJC12 deficiency in the literature [13,14]. There is a need to identify a reliable and non-invasive biomarker to guide treatment initiation and monitoring of DNAJC12 deficiency. Therefore, until clear guidelines for treatment of DNAJC12-deficiency patients are available, physicians should not rely on any single biomarker to monitor treatment response. Instead, both clinical and biochemical findings should be collectively considered in guiding the management of DNAJC12-deficiency patients.

## 4. Conclusions

Here, we report the first case of DNAJC12 deficiency diagnosed in Hong Kong, picked up by HPA through newborn screening. A novel variant in the *DNAJC12* gene was identified. This patient highlights the importance of including this novel differential diagnosis in the workup of HPA, as early treatment may lead to better clinical outcomes. However, the understanding of this disease remains limited, with cases reported in the literature varying widely in severity. More international collaborative efforts are needed to study the prevalence, clinical course, long-term outcomes, and optimal treatment and monitoring strategies for DNAJC12 deficiency.

## Figures and Tables

**Table 1 IJNS-10-00074-t001:** Biochemical parameters monitored during the disease course.

Age	Blood Phe (umol/L) *	Blood Prolactin (mIU/L) RR 3 Months Onwards 49–280 mIU/L	CSF HVA (nmol/L) RR 295–932 nmol/L	CSF 5-HIAA (nmol/L) RR 114–336 nmol/L
Day 2	233	N/A	N/A	N/A
Day 12	272	2526 RR 78–1721 mIU/L	N/A	N/A
3 months	284	957	248	75
4 months (start of treatment)	351	781	N/A	N/A
12 months	429	1349	N/A	N/A
16 months	589	929	N/A	N/A
21 months	567	1268	369	137
24 months	189	395	N/A	N/A
3 years	250	323	N/A	N/A
4 years	213	588	N/A	N/A
4 years 3 months	554	344	N/A	N/A
4 years 6 months	231	652	N/A	N/A

Abbreviations: 5-HIAA, 5-hydroxyindolacetic acid; DBS, dried blood spot; HVA, homovanillic acid; N/A, not available; Phe, phenylalanine; RR, Reference range. * All of the samples were plasma samples, except for the day 2 sample, which was a dried blood-spot sample from newborn screening.

**Table 2 IJNS-10-00074-t002:** Pathogenic and likely pathogenic variants in Chinese patients with DNAJC12 deficiency.

Variants	Inheritance	Nucleotide Change	Amino Acid Change	Variant Type
Current report	Compound heterozygous	c.185C>A *	p.Ala62Glu	Missense
		c.214C>T	p.Arg72Ter	Nonsense
Li et al. [12]	Compound heterozygous	c.306C>G	p.His102Gln	Missense
		c.182delA	p.Lys61ArgfsTer6	Frameshift
Wang et al. [20]	Compound heterozygous	c.158-1G>A	p.Val53AspfsTer15	Frameshift
		c.336delG	p.Met112IlefsTer44	Frameshift
Feng et al. [21]	Homozygous	c.262del	p.Gln88SerfsTer6	Frameshift

Abbreviations: ACMG, American College of Medical Genetics and Genomics; AMP, Association for Molecular Pathology. * Novel variant not previously reported in population control exomes and genomes.

## Data Availability

Data are contained within the article.

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
