# Peer review of "DNAJC12 Deficiency, an Emerging Condition Picked Up by Newborn Screening: A Case Illustration and a Novel Variant Identified"

_2409-515X, 2024, doi:10.3390/ijns10040074_

Round 1
Reviewer 1 Report
Comments and Suggestions for Authors
See attachment

Minor edits required
Author Response
L87: “urine metabolic profile” is used when the method used appears to be specifically targeting urine pterins. Please rename the test to avoid confusion with other tests with a similar name.
Thank you for your comment, “urine metabolic profile” has been changed to “urine pterins” in the text.
L91: Sanger sequencing, capital S
Thank you for pointing this out, the word Sanger has been capitalized.
L106: Describing the novel missense variant as “disease-causing” is presumptive when it has not been reported previously. Please reword to make it clear that only the nonsense variant has been previously reported. Also, a lit reference to the nonsense variant should be given here. Which databases was the missense variant absent from? Please list.
Thank you for your comment. The paragraph regarding the two variants identified in the patient has been modified to describe the variants more accurately and in greater detail.
At three months of age, he also underwent targeted analysis of the whole DNAJC12 gene (NM_021800.3) by Sanger sequencing. All the coding exons and the flanking introns (+/- 40bp) of the gene were analyzed. He was found to be compound heterozygous for two variants, NM_021800.3: c.214C>T p. (Arg72*) and NM_021800.3: c.185C>A p. (Ala62Glu). Parental genotyping showed that the nonsense variant is inherited from the father while the missense variant is inherited from the mother.
The nonsense variant, c.214C>T, replaced an arginine with a termination codon at position 72 of the DNAJC12 protein. This variant is a known pathogenic variant that has been reported in multiple patients with DNAJC12 deficiency in homozygous state [15].
The missense variant, c.185C>A, is a novel variant that is absent in control database (gnomAD exomes and genomes v2.1.1). The REVEL score was 0.808, indicating moderate pathogenicity. The clinical presentations of isolated hyperphenylalaninemia, normal urine and CSF pterin patterns, and CSF findings of reduced 5-HIAA and HVA are highly suggestive of DNAJC12 deficiency. Therefore, the missense variant was curated as likely pathogenic according to the American College of Medical Genetics and Genomics (ACMG) standards and guidelines for the interpretation of sequence variants [16-19].
L216: How do the authors know there have not been any cases missed by NBS? What was the surveillance? Did they consult all relevant metabolic/neuro etc clinics to check? Please elaborate.
Thank you, it is a valid point that we can’t be 100% certain that we have not missed any cases by NBS. The surveillance is by voluntary reporting from paediatricians and physicians in Hong Kong. Given that there is just a small community of paediatrics/adult neurologists and metabolic physicians, we have presented our case in relevant societies and have yet to receive feedback of missed cases. We understand that this does not rule out false negative cases. However, the message we are trying to convey in this paragraph is that the possibility of DNAJC12 deficiency should not be overlooked even with a normal NBS result. So, if there are false negative cases in Hong Kong that have not been reported to the NBS team, this message remains valid or is even more substantiated.
We have modified the sentence as follows to allow readers to know the limitation of our surveillance system:
We have not encountered any false negative cases till date in our NBS program in Hong Kong, where the surveillance for false negative cases requires voluntary reporting from paediatrics specialists and physicians.
Reviewer 2 Report
Comments and Suggestions for Authors
The manuscript is a case report of a child with hyperphenylalaninemia (HPA) detected by newborn screening. Clinical workup eventually determined the child carried one likely pathogenic variant and one pathogenic variant in DNAJC12, in trans. The case highlights the need for clinicians to consider DNAJC12 deficiency in babies referred for workup following NBS identification of HPA. The likely pathogenic variant discovered has not been previously reported and is a contribution to the medical genetics field. In addition, this manuscript combined with the previous case reports of DNAJC12 deficiency in HPA patients serves to remind clinicians to consider DNAJC12 sequencing in unresolved cases. The paper is well written and thorough.
Minor points
Mutation vs variant. Consider replacing ‘mutation’ throughout with ‘pathogenic/likely pathogenic variant’. Current guidelines discourage the use of ‘mutation’. See https://hgvs-nomenclature.org/stable/background/basics/#mutation-and-polymorphism and Richards 2015, Genet.Med. 17:405-424
Line 104: should NM_021800.2 be NM_021800.3 as it is for the annotated variants?
Line 109: List ACMG scores that contributed to the LP classification.
Line 176: ‘vairant’ should be ‘variant’
Line 196: only three of 20. The fourth only carried one DNAJC12 variant. Also, “were carriers of the PAH gene” should be corrected to “were carriers of a pathogenic variant in the PAH gene”
Line 216: NBS acronym was already introduced.
Line 249: ASD acronym was already introduced.
Figure 1: I’m curious how this figure fits with the proposed ACMG algorithm for NBS results of elevated Phe (https://www.ncbi.nlm.nih.gov/books/NBK55827/bin/Phenylalanine-Algorithm.pdf). Would your figure apply directly above the ‘Molecular genetic testing’ box? Or is your algorithm an alternative? Also, since there are reports of asymptomatic patients who carry two pathogenic variants in DNAJC12, would detection of variants be enough to warrant diagnosis of DNAJC12 deficiency? Or would the detected variants be correlated with clinical findings? It might be outside of the scope of this manuscript, but with so many case reports of patients with variants detected, it could be helpful to define what clinical findings warrant (e.g. CSF neurotransmitters) treatment.
Author Response
Consider replacing ‘mutation’ throughout with ‘pathogenic/likely pathogenic variant’. Current guidelines discourage the use of ‘mutation’.
Thank you for the comment. We have amended the words “mutation/mutations” to “variant/variants.”
Line 104: Should NM_021800.2 be NM_021800.3 as it is for the annotated variants?
Yes, thank you for pointing this out. We have amended it to NM_021800.3.
Line 109: List ACMG scores that contributed to the LP classification.
Thank you for your comment. The paragraph regarding the two variants identified in the patient has been modified to describe the variants more accurately and in greater detail.
At three months of age, he also underwent targeted analysis of the whole DNAJC12 gene (NM_021800.3) by Sanger sequencing. All the coding exons and the flanking introns (+/- 40bp) of the gene were analyzed. He was found to be compound heterozygous for two variants, NM_021800.3: c.214C>T p. (Arg72*) and NM_021800.3: c.185C>A p. (Ala62Glu). Parental genotyping showed that the nonsense variant is inherited from the father while the missense variant is inherited from the mother.
The nonsense variant, c.214C>T, replaced an arginine with a termination codon at position 72 of the DNAJC12 protein. This variant is a known pathogenic variant that has been reported in multiple patients with DNAJC12 deficiency in homozygous state [15].
The missense variant, c.185C>A, is a novel variant that is absent in control database (gnomAD exomes and genomes v2.1.1). The REVEL score was 0.808, indicating moderate pathogenicity. The clinical presentations of isolated hyperphenylalaninemia, normal urine and CSF pterin patterns, and CSF findings of reduced 5-HIAA and HVA are highly suggestive of DNAJC12 deficiency. Therefore, the missense variant was curated as likely pathogenic according to the American College of Medical Genetics and Genomics (ACMG) standards and guidelines for the interpretation of sequence variants [16-19].
Line 176: ‘vairant’ should be ‘variant’
Thank you and we apologize for the typo.
Line 196: only three of 20. The fourth only carried one DNAJC12 variant. Also, “were carriers of the PAH gene” should be corrected to “were carriers of a pathogenic variant in the PAH gene”
Thank you for the comment, we have amended the sentence accordingly:
Moreover, three of these 20 cases were carriers of a pathogenic variant in the PAH gene but were previously misdiagnosed to have PAH deficiency.
Line 216: NBS acronym was already introduced/ Line 249: ASD acronym was already introduced.
Thank you for the comment, the text has been adjusted accordingly.
Figure 1: I’m curious how this figure fits with the proposed ACMG algorithm for NBS results of elevated Phe (https://www.ncbi.nlm.nih.gov/books/NBK55827/bin/Phenylalanine-Algorithm.pdf). Would your figure apply directly above the ‘Molecular genetic testing’ box? Or is your algorithm an alternative? Also, since there are reports of asymptomatic patients who carry two pathogenic variants in DNAJC12, would detection of variants be enough to warrant diagnosis of DNAJC12 deficiency? Or would the detected variants be correlated with clinical findings? It might be outside of the scope of this manuscript, but with so many case reports of patients with variants detected, it could be helpful to define what clinical findings warrant (e.g. CSF neurotransmitters) treatment.
Thanks for your comment regarding this flow chart. Considering the availability of the proposed ACMG algorithm and another flow chart for hyperphenylalaninemia suggested by Blau et al. in their article titled “DNAJC12 deficiency: A new strategy in the diagnosis of hyperphenylalaninemia,” we have decided to remove the algorithm from our manuscript to avoid duplication and confusion.
Reviewer 3 Report
Comments and Suggestions for Authors
The manuscript “DNAJC12 deficiency: An emerging condition picked up by newborn screening; a case illustration and a novel variant identified” focus on an actual subject for those working in the field. It is well written and more than reporting a new DNAJC12 deficiency case detected by NBS, it includes a brief review on the subject. It fits on the scope of IJNS.
Nevertheless, some minor revisions are needed:
· Line 35: Please clarify the meaning of the sentence “Although tandem mass spectrometry is used to measure Phe concentrations nowadays, an elevated Phe (hyperphenylalaninemia (HPA)) remains a core primary analyte to initiate call backs in majority of newborn screening programs worldwide”. Phe is far from being the main cause of recall in MS/MS NBS.
· Line 69: Blau et al 2018 (ref 19 of the manuscript) proposed a flow-chart for the diagnosis of DNACJ12 deficiency. Please revise the sentence on lines 69/70 accordingly.
· Some formation issues must be revised (ex. Insert a space between lines 136 and 137 and between lines 175 and 176).
· Lines 196/197: “Moreover, four of these 20 cases were carriers of the PAH gene but were previously misdiagnosed to have PAH deficiency [13]”. What means “carriers of the PAH gene“? Were these four cases carriers of PAH variants/mutations? Please clarify.
· Figure 1:
o On the second box – “Pre-meal: DBS, PAA, PRL”. DBS is a sample matrix. Please clarify what should be tested on DBS.
o On this flow-chart are excluded the analysis of Pterins and DHPR on DBS – Please justify this option.
o Please include on the flow chart what happens after DNAJC12 gene analysis. What conclusions can we get after DNAJC12 gene analysis and how to proceed?
Author Response
Line 35: Please clarify the meaning of the sentence “Although tandem mass spectrometry is used to measure Phe concentrations nowadays, an elevated Phe (hyperphenylalaninemia (HPA)) remains a core primary analyte to initiate call backs in majority of newborn screening programs worldwide”. Phe is far from being the main cause of recall in MS/MS NBS.
Thank you for the comment, the purpose of this sentence was to emphasize that despite the widespread use of tandem mass spectrometry, which allowed the analysis of numerous analytes simultaneously, Phe remains one of the primary analytes that is being looked at in newborn screening. We have rephrased the sentence as follow:
Although tandem mass spectrometry has become the mainstay of newborn screening nowadays and allows the analysis of multiple analytes, an elevated Phe (hyperphenylalaninemia (HPA)) remains one of the primary analytes to initiate call backs in newborn screening programs worldwide.
Line 69: Blau et al 2018 (ref 19 of the manuscript) proposed a flow-chart for the diagnosis of DNACJ12 deficiency. Please revise the sentence on lines 69/70 accordingly.
Thank you for the comment. We have revised the sentence to focus on the management and monitoring instead of diagnosis of DNAJC12 deficiency. We have also modified line 204-205 to point out that Blau et al has proposed a diagnostic flow-chart in their report.
Also, there is no established consensus on the treatment and monitoring of DNAJC12 deficiency. Current treatment strategies are largely based on the consensus guideline published for disorders of BH4 deficiencies.
Blau et al. emphasized the importance of ruling out DNAJC12 deficiency in patients with unexplained HPA, and proposed a diagnostic flow-chart in their report [19].
Some formation issues must be revised (ex. Insert a space between lines 136 and 137 and between lines 175 and 176).
Thank you for the comment. A space has been added between the mentioned lines.
Lines 196/197: “Moreover, four of these 20 cases were carriers of the PAH gene but were previously misdiagnosed to have PAH deficiency [13]”. What means “carriers of the PAH gene“? Were these four cases carriers of PAH variants/mutations? Please clarify.
Thank you for the comment. The sentence is amended as follows after reviewing the paper:
Moreover, three of these 20 cases were carriers of a pathogenic variant in the PAH gene but were previously misdiagnosed to have PAH deficiency.
Figure 1:
o On the second box – “Pre-meal: DBS, PAA, PRL”. DBS is a sample matrix. Please clarify what should be tested on DBS.
o On this flow-chart are excluded the analysis of Pterins and DHPR on DBS – Please justify this option.
o Please include on the flow chart what happens after DNAJC12 gene analysis. What conclusions can we get after DNAJC12 gene analysis and how to proceed?
Thanks for your comment regarding this flow chart. Considering the availability of the proposed ACMG algorithm and another flow chart for hyperphenylalaninemia suggested by Blau et al. in their article titled “DNAJC12 deficiency: A new strategy in the diagnosis of hyperphenylalaninemia,” we have decided to remove the algorithm from our manuscript to avoid duplication and confusion.
Reviewer 4 Report
Comments and Suggestions for Authors
Case report 1
DNAJC12 deficiency: An emerging condition picked up by newborn screening; a case illustration and a novel variant identified
The paper presents a case of DNAJC12 deficiency diagnosed in Hong Kong, detected through elevated phenylalanine (Phe) levels and an increased Phe/Tyr ratio during newborn screening. The association between DNAJC12 deficiency and hyperphenylalaninemia has been known since 2017,and in this context the paper is not novel. However this work contributes by reporting the first case in Hong Kong and identifying both a novel missense change and a pathogenic nonsense variant.
The authors should consider revising the diagnostic algorithm proposed in Figure 1. With the advancements in Next-Generation Sequencing (NGS) technologies, genetic testing can now cover a broad spectrum of genes linked to hyperphenylalaninemia, including DNAJC12. Incorporating DNAJC12 analysis simultaneously with other relevant genes would accelerate the identification of the deficient gene, offering significant advantages for the patient's clinical management and care by saving valuable time in diagnosis.
Comments on the Quality of English Languageenglish must be revised
Author Response
The authors should consider revising the diagnostic algorithm proposed in Figure 1. With the advancements in Next-Generation Sequencing (NGS) technologies, genetic testing can now cover a broad spectrum of genes linked to hyperphenylalaninemia, including DNAJC12. Incorporating DNAJC12 analysis simultaneously with other relevant genes would accelerate the identification of the deficient gene, offering significant advantages for the patient's clinical management and care by saving valuable time in diagnosis.
Thanks for your comment regarding this flow chart and we are aware that resource availability in different countries or regions would certainly affect the diagnostic pathway. For example, an NGS panel can directly look at relevant genes related to hyperphenylalaninemia, without having to rule out PAH deficiency before moving on the DNAJC12 deficiency.
Considering the availability of the proposed ACMG algorithm and another flow chart for hyperphenylalaninemia suggested by Blau et al. in their article titled “DNAJC12 deficiency: A new strategy in the diagnosis of hyperphenylalaninemia,” we have decided to remove the algorithm from our manuscript to avoid duplication and confusion.
Extensive editing of English language required:
Thank you for your comment. Amendments to wordings and sentence structures have been made, highlighted in green, to make the delivery of our messages more concise.
Round 2
Reviewer 4 Report
Comments and Suggestions for Authors
authors replyed to all my comments. the manuscript can be accepted in present form